

# Bias Corrections of Precipitation Measurements across Experimental Sites in Different Ecoclimatic Regions of Western Canada

Xicai Pan[1], Daqing Yang[2], Yanping Li[1], Alan Barr[2], Warren Helgason[1], Masaki Hayashi[3], Philip Marsh[4], John Pomeroy[5], Richard J. Janowicz[6]

[1]Global Institute for Water Security, University of Saskatchewan, 11 Innovation Boulevard, Saskatoon, Canada

[2]National Hydrology Research Centre, Environment Canada, 11 Innovation Boulevard, Saskatoon, Canada

[3]Department of Geoscience, University of Calgary, Calgary, Alberta, Canada

[4]Cold Regions Research Centre, Wilfrid Laurier University, Waterloo, Ontario, Canada

[5]Centre for Hydrology, University of Saskatchewan, 117 Science Place, Saskatoon, Canada

[6]Water Resources Branch, Yukon Department of Environment, Whitehorse, Yukon, Canada

*Correspondence to*: Y. Li (yanping.li@usask.ca)

**Abstract.** This study assesses a filtering procedure on accumulating precipitation gauge measurements, and quantifies the effects of bias corrections for wind-induced undercatch across four ecoclimatic regions in western Canada, including the permafrost regions of the Sub-arctic, the Western Cordillera, the Boreal Forest, and the Prairies. The bias corrections increased monthly precipitation by up to 163% at windy sites with short vegetation, and sometimes modified the seasonal precipitation regime, whereas the increases were less than 13% at sites shielded by forest. On a yearly basis, the increase of total precipitation ranged from 8 to 20 mm (3-4%) at sites shielded by vegetation, and 60 to 384 mm (about 15-34%) at open sites. In addition, the bias corrections altered the seasonal precipitation patterns at some windy sites with high snow percentage (>50%). This study highlights the need and importance of precipitation bias corrections at both research sites and operational networks for water balance assessment and the validation of global/regional climate/hydrology models.

## 1 Introduction

Accurate precipitation data are essential for understanding climate change and associated hydrological responses from small basins to large regions around the world. However, measuring precipitation particularly snowfall in cold regions is still difficult. The quality of precipitation measurements is commonly affected by the limitations of the precipitation gauges and by gauge setting and shielding. Precipitation gauge technology has improved significantly in the last century. The most widely used precipitation gauges are classified into four types: manual gauges, tipping-bucket rain gauges (TBRG), weighing type gauges, and optical gauges. Each gauge type has advantages and disadvantages. For instance, the TBRG performs well for liquid precipitation, while the weighing gauge can measure both liquid and solid precipitation in most weather conditions. Depending on site characteristics and environment, gauge performance can vary widely. Results from numerous studies show that gauge type and collection method significantly affect measurement precision and accuracy (Emerson and Macek-Rowland, 1990; Yang et al., 1999a). Although all precipitation measurements are prone to bias, the measurement biases are most serious in cold regions due to high





snowfall percentage (Goodison et al., 1998; Yang et al., 1998; Yang et al., 1999b). Corrections for systematic biases in gauge measurements, such as wind-induced undercatch, wetting loss, evaporation loss, and underestimation of trace precipitation amounts are necessary (Goodison et al., 1998).

Currently the Geonor T200-B series accumulating gauge is widely used in many nations including USA and Canada. Since this type of gauge, through proper implementation and maintenance (i.e. providing sufficient oil or antifreeze in the collecting bucket), can prevent excessive evaporation losses, biases due to trace events, wetting loss and evaporation loss are relatively small in comparison to the wind-induced errors (Goodison et al., 1998; GEONOR, 2012). In addition, a variety of artefacts (noise) associated with high frequency precipitation data have been observed (Baker et al., 2005; Lamb and Swenson, 2005; Fortin et al., 2008). Properly detecting and excluding noises from the high time resolution measurements are essential to quality control and data processing. Automatic precipitation gauges have been used at both network stations and research sites. Relative to national weather stations, the measurement issues are often greater in research networks, where the installations may be in harsh environments and suffer from irregular maintenance. As such, significant effort is required to investigate the quality of precipitation measurements from research sites in the northern regions. Careful analysis of Geonor gauge data collected in different regions will lead to a better understanding of automatic gauge performance and observation biases across various environments.

Corrections for wind-induced gauge undercatch are of great interest for regional climate and hydrology studies. Previous studies (e.g. Yang et al., 1998; Adam and Lettenmaier, 2003) carried out the bias corrections for the national standard (manual) gauges on a daily time scale. This study focused on sub-daily time-scale precipitation data measured by automatic precipitation gauges and applies an hourly relationship between wind speed and Geonor gauge catch efficiency. Based on a proper data quality control through noise filtering of gauge signals, we quantified the magnitude of the bias corrections at seven selected research sites in western Canada. Specifically, the objectives of this study are: (1) to compare the Geonor gauge rainfall data with the TBRG records so as to assess a filtering procedure for different levels of noises; (2) to correct wind-induced undercatch for hourly Geonor gauge measurements; and 3) to compare the magnitude of the bias corrections and to assess their effects to precipitation regimes across different ecoclimatic regions. The methods and results of this study will have a significant impact on regional climate analyses, water balance calculations and validations of regional climate/hydrology models.

## 2 Materials and Methods

### 2.1 Sites and data

Seven research sites were selected from the Changing Cold Regions Network (CCRN) (DeBeer et al., 2015) over the cold interior of western Canada (Fig. 1). These monitoring stations cover four ecoclimatic regions: the permafrost regions of the Sub-arctic, the Western Cordillera, the Boreal Forest, and the Prairies. A brief overview of the sites and data from north to south is given as follows.

The Trail Valley Creek site (TVC) is located 50 km north-northeast of Inuvik, Northwest Territories, in the continuous permafrost zone. The basin is dominated by gently rolling hills and some deeply incised river valleys. The



upland tundra area is vegetated with grasses, lichens, and mosses, while shrub tundra occupies the moister hillslopes and valley bottoms with vegetation height ranging from 0.5 m to 3 m. The climate is characterized by short summers and long cold winters with an 8-month snow-cover period. Large snowdrifts form in winter (Pomeroy et al., 1997), and about half of the annual total precipitation (a mean of 231 mm over 1991-2000) is in solid form (Marsh et al., 2004).

The Buckbrush/Wolf Creek (WCB) and Forest/Wolf Creek (WCF) sites are located in the Wolf Creek Research Basin, Yukon Territory within the Upper Yukon River Basin. They are representative of the interior Sub-arctic Cordilleran landscape. The typical mountain environment includes dense boreal forest at lower elevations, sparse forest, open meadow and shrub tundra at the higher elevations, and exposed alpine areas with mostly bare rock at the highest elevations. The WCB and WCF are situated at two different areas of tall shrub tundra (Pomeroy et al., 2006)
and mature white spruce forest (Pomeroy et al., 2002), respectively. The sub-arctic continental climate is characterized by a large variation in air temperature, low relative humidity and relatively low precipitation (Wahl et al, 1987). The monthly mean air temperature ranges from -20°C in winter to 15°C in summer, and the mean annual precipitation is 300 to 400 mm, with approximately 40% as snowfall (Janowicz et al., 2004; Rasouli et al., 2014). The used forest gauges were installed in a natural clearing of approximately 15 m diameter within a dense, mature white spruce forest.
The used shrub tundra gauges were installed with its orifice above the shrub canopy.

The Old Jack Pine/Boreal Ecosystem Research and Monitoring Sites (BERMS) site is located in the southern Boreal Forest within mid-Boreal upland and Boreal Transition eco-regions. The gently rolling landscape is covered by mature jack pine, black spruce, aspen and mixed wood forest, and wetland fen. About one quarter of the annual total precipitation (about 500 mm) is received as snow, and the mean annual air temperature is around 0.4°C (as observed at
Waskesiu Lake over 1971-2000, Barr et al., 2009). The gauges were installed in a natural clearing of approximately 20-30 m diameter, surrounded by 14m-tall jack pine trees.

The Brightwater Creek site (BC) is situated in a grazing pasture surrounded by flat agricultural land in the Canadian Prairie region. Solid precipitation commonly occurs between November and April, constituting about one third of the total annual precipitation (~400 mm). The average snowpack typically develops to a depth of 15-30 cm, and blowing
snow frequently occurs (Pomeroy et al., 1993). The monthly mean air temperature ranges from -12.9°C in January and to 18.8°C in July.

The Western Nose Creek site (WNC) is located at the western edge of the Canadian Prairies. The landscape is similar to BC, but mid-winter snowmelt often occurs (Mohammed et al., 2013). At the Calgary International Airport, located 20 km southeast of the site, 1981-2010 normal annual precipitation is 482 mm and the monthly mean air
temperatures in January and July are -7.1°C and 16.5°C, respectively (Hayashi and Farrow, 2014).

The Fisera Ridge/Marmot Creek site (MC) is located in the Rocky Mountain Front Ranges on a larch covered ridge top at an elevation of 2325 m. The gauge is located on a 3 m tall pedestal in a 10 m diameter clearing in the 10 m tall larch forest. The mountains range in elevation from 1600 m to 2800 m, and typical mountain environments include montane and subalpine forest cover, alpine tundra and talus/rock at higher elevation. The climate is characterized by
cool wet summers and long cold winters, and monthly mean air temperature ranges from -10.7°C in January to 11.7°C





in July (Pomeroy et al., 2012; Fang et al., 2013; Harder et al., 2015). There is a significant difference in the annual precipitation from 638 mm at the valley bottom to 1100 mm at higher elevations (Storr, 1967).

Meteorological variables including air temperature, relative humidity, wind speed, rainfall and cumulative total precipitation, were measured at all sites with an interval of 30 minutes except the MC site with an interval of 15 minutes. Sensor information and data periods are listed in Table 1. Two types of precipitation gauges were used at the sites: a TBRG and the Geonor T200-B accumulating gauge, as shown in Fig. 2. The TBRG funnels rain into a mechanical device which tips when it has collected the equivalent of 0.1 mm of rainfall (amount varies in bucket size). Three types of TBRG were deployed across the seven sites (Table 1): Hydrological Services tipping bucket rain gauge (Model TB4, Hyquest Solutions PTY LTD); Meteorological Service of Canada tipping bucket; and the TE525MM gauge (Texas Electronics) were used at these sites. The Geonor T200-B gauge was deployed in a standard configuration with a (single) Alter shield (which we will denote Geonor-SA) to increase the snowfall catch efficiency (Fig. 2b). The measuring mechanism of the Geonor T200-B uses one or more vibrating wires to continuously weigh the content in the bucket. It can record changes with a resolution of 0.1 mm at the time scale up to minutes. Oil was added to the Geonor gauge container to minimize losses by evaporation.

## 2.2 Algorithms for data processing

The output from the Geonor accumulating gauge was post-processed: to identify individual precipitation events; and to correct for long-term drift associated with sublimation and evaporation. Diurnal (or a bit longer) noise often occurs at sites with strong diurnal changes in temperature, radiation and wind speed. Long-term drift results from evaporation from inside the bucket (Duchon, 2008). A general algorithm for data processing was applied in two steps: (1) manual pre-filtering to remove obvious outliers and eliminate changes associated with gauge servicing, e.g. emptying and/or adding of antifreeze and oil; (2) automated filtering to eliminate the above mentioned two types of noises. Step 2 used a "brute-force" filtering algorithm to eliminate negative and small positive changes by combining them with proximate, positive changes above a specified threshold. The changes were aggregated beginning with the most negative change in the time series then continuing until all changes below the threshold were eliminated. In this study, the threshold was set to 0.1 mm, the precision of the Geonor T200-B gauge. The "brute-force" filter preserves the total precipitation accumulation but aggregates all changes into values above the threshold. The filter will thus fail during periods with visually evident declines in the time series. When negative drift was observed, its impact was minimized by applying the filter over a moving window of one or a few days. The window size is related to the drift period. All filtering was visually supervised to confirm that it performed reasonably.

## 2.3 Correcting for Geonor-SA undercatch

After filtering, 30 min precipitation data were produced for all the sites with different observational time periods. With these data, a bias correction was applied to solid precipitation, using an empirical relationship between catch efficiency and wind speed derived in Canada (Smith, 2007)





$$P_{\text{corr}} = P_{\text{obs}} / CE , \quad CE = 1.18 e^{-0.18 W_s} ,$$ (1)

where $P_{\text{corr}}$ (mm) is the corrected precipitation, $P_{\text{obs}}$ (mm) is the measured solid precipitation after filtering, catch efficiency $CE$ is the ratio of the Geonor catch to the "true" snowfall measured by a WMO reference called the double fence intercomparison reference (DFIR) (Goodison et al., 1998), and $W_s$ (m s$^{-1}$) is the hourly mean wind speed at the gauge height. Wind-induced bias corrections are also necessary for rainfall, though they are not as significant as for snowfall (Goodison et al., 1998; Yang 1998b). An investigation of rainfall measurements at Egbert, Ontario (Devine and Mekis, 2008) showed a catch ratio of 95% for the Geonor-SA relative to a pit gauge, the WMO reference for rainfall intercomparison (WMO, 1969). Thus, we adjusted all rainfall measurements using the average catch efficiency of 95% ($CE = 0.95$).

Over-correction is possible for snowfall events due to the impact of blowing snow at high wind speeds. To avoid over-correction, an upper threshold wind speed is required (Goodison et al., 1998). For daily precipitation totals, an upper threshold for daily mean wind speed of 6.5 m s$^{-1}$ is often applied in Arctic and northern regions (Yang et al., 2005). In this study, we used lower and upper threshold $W_s$ of 1.2 and 9.0 m s$^{-1}$, respectively. Thus, $CE$ was set to 1.0 when $W_s$ < 1.2 m s$^{-1}$, and to 0.23 ($CE$ for 9 m s$^{-1}$ wind speed) when $W_s$ > 9.0 m s$^{-1}$. Smith (2007) tested this relationship at an open and windy site near Regina and concluded the method was generally applicable for the interior of western Canada or other regions.

The bias correction algorithm for precipitation measurements requires the determination of precipitation types. A variety of approaches for precipitation phase determination has been summarized in Harder and Pomeroy (2013). Commonly, a double temperature threshold is used to distinguish snowfall, rainfall and mixed events. All precipitation below the lower threshold or above the upper threshold is considered as snow or rain, respectively, and between the defined thresholds is then considered to be mixed with certain proportion (Pipes and Quick, 1977). With a solid physical basis, a threshold of hydrometeor temperature ($T_i$), approximating the temperature at the surface of a falling hydrometeor is more robust than others directly using air temperature ($T_a$) or dew point temperature ($T_d$) (Harder and Pomeroy, 2013). The $T_i$ can be derived from near-surface meteorological variables including air temperature and humidity by using the psychrometric energy balance method (Harder and Pomeroy, 2013)

$$T_i = T_a + \frac{D}{\lambda_t} L (\rho_{Ta} - \rho_{\text{sat}(Ti)}) ,$$ (2)

corresponding parameters for Eq. (2) are listed in Table 2. An empirical relationship between $T_i$ and the rainfall fraction is then applied to separate snowfall and rainfall in precipitation measurements,

$$f_r (T_i) = \frac{1}{1 + b \times c^{T_i}} ,$$ (3)

where the calibrated coefficients $b = 2.630006$ and $c = 0.09336$ are based on the measurements (15 min time interval)



in a small Canadian Rockies catchment, Marmot Creek Research Basin ( Harder and Pomeroy, 2013).

Wind speed at gauge height is necessary to determine gauge undercatch. When wind speed was not measured at gauge height, it was estimated from wind field profile models over different canopies. For open site with short grass (e.g. BC and WNC), a logarithmic model is employed as

$$W_s(h) = W_s(H) \left[ \frac{\ln(h/z_0)}{\ln(H/z_0)} \right],$$

(4)

where $W_s(h)$ is the estimated hourly mean wind speed at the gauge height, (m s$^{-1}$), $W_s(H)$ is the measured hourly mean wind speed at the available height, (m s$^{-1}$), $H$ is the height of the available anemometer, (m), and $z_0$ is the roughness length, (m), set to 0.01 m for the cold period for snow surface ($T_a \leq 0°C$), and 0.03 m for short grass in the warm period ($T_a > 0°C$) (Yang et al., 1998b). For sites with shrub or forest canopies (e.g. WCB and WCF), wind speed within canopy is calculated from an exponential wind profile model (Cionco, 1965)

$$W_S(h) = W_S(H) \exp(\alpha(h/h_0 - 1))$$

(5)

where $h_0$ is the average canopy height (m), $W_s(H)$ is the mean wind speed at the canopy height, and $\alpha$ is the canopy flow index. The wind speed measurements above canopy at WCB and WCF are approximately assumed at the canopy height, and the $\alpha$ is set as 1.7 for shrub and forest based on suggested values for similar canopies (e.g. Raupach et al., 1996; Wang and Cionco, 2007). In addition, the wind speed measurements at BERMS is assumed as at gauge height due to the small height difference.

## 3 Results

### 3.1 Geonor-SA vs. TBRG for rain

Measurement noise in this study can be grouped into two classes: (1) irregular diurnal or longer drift, depending on the time scale of temperature or wind speed fluctuations; and (2) evident declines in the accumulation due to evaporation losses. Example of the two typical types of noise in Geonor precipitation measurements and corresponding filtering are shown in Fig. 3. The former occurs at all sites, with a dynamic range of ±0.1 mm at the forest sites, e.g. BERMS, to ±3 mm at the grassland sites, e.g. BC. This type of noise is usually larger in cold periods than in warm seasons, and can be related to turbulent pressure fluctuations (i.e. wind pumping), as well as diurnal temperature effects on gauge transducers. The second type of noise, evaporation losses, occurred at most sites. However, significant declines are mainly observed at MC.

Relative to TBRG observations, the filtering of Geonor observations may produce artifacts, erroneously creating or removing light rainfall events. The effect of filtering on Geonor precipitation measurements can be assessed by comparing the filtered rainfall from the Geonor-SA ($P_g$) with the TBRG measurements ($P_t$) (Fig. 4). Overall, the rainfall measurements by the two gauges (the blue dots around the 1:1 line) agree closely at all sites. A few notable





differences like in Fig. 4k is attributed to poor performance of the TBRG at WNC during some heavy events, where over-measurement can be deduced by comparison with the Geonor measurements. In addition, the relatively small differences might be attributed to different catch efficiencies.

Two special cases for "missing" measurements, with: (1) $P_t > 0$, $P_g = 0$; and (2) $P_t = 0$, $P_g > 0$, are marked with red and black circles in the left panels of Fig. 4. Generally, most "missing" measurements of heavy rainfall (e.g. > 5 mm/30min) are related to a false record in both gauges, however their occurrence is rare. In contrast, "missing" measurements of light rainfall occur frequently, related to the artifacts of filtering. In situations of widely fluctuating noise, since "brute-force" filter may introduce anomalous rainfall events or remove real events (e.g. Fig. 3a), without adding bias to the total accumulation, the annual totals of "missing" $P_t$ in case (1) and missing $P_g$ in case (2) should be approximately equal.

Yearly totals of "missing" $P_t$ and $P_g$ are shown in the right panel in Fig. 4. For example, BC (Fig. 4j) has yearly totals of "missing" $P_t$ from 12 to 48 mm and $P_g$ from 16 to 46 mm, respectively. The absolute offsets are all less than 7 mm except in 2010. Through comparing the "missing" measurements of $P_t$ and $P_g$, we have found t most events are light precipitation (< 1 mm) and are introduced by the $P_g$ filtering. Light precipitation events can be masked by diurnal fluctuations of noise, which can be removed by the filter. On the other hand, the filter may anomalously create light precipitation events near the start of actual rain events. For yearly totals, the two artifacts of created and removed rainfall events offset each other and do not introduce significant bias. In contrast, the notable imbalance between $P_t$ and $P_g$ in 2010 is related to the obvious "missing" measurements of $P_t$, which are likely caused by false TBRG readings, related to TBRG plugging. Similar results can be found at TVC. But the yearly totals of "missing" $P_t$ and $P_g$ except 2013 are smaller (< 15 mm/yr) because of lower noise and fewer rainfall events at TVC than BC. The site WNC has particularly large imbalances between $P_t$ and $P_g$, which is due to false Geonor and TBRG readings. In addition, hailstorms in this region may also contribute to the imbalances.

Overall, we conclude that the filtering algorithm works well at the study sites, although it sometimes removes and creates some light precipitation events (case (1) and case (2)). The number of these two artifacts is influenced by noise status and precipitation characteristics. However, the two artifacts typically offset each other so that the net effect is usually small (< 20 mm/yr) for rainfall data).

### 3.2 Monthly precipitation and bias-corrections

Based on 30-min data and bias corrections, Fig. 5 summarizes the bias correction for wind-induced undercatch at monthly time scales. The left panels compare the monthly mean values of measured and corrected precipitation over the period 2006-2015; the corresponding meteorological summaries are shown in the right panels. Generally, the effects of the bias corrections on precipitation measurements at the study sites are controlled by their ecoclimatic characteristics. Here the results are described from north to the south across the study region.

For TVC, the results show monthly measured precipitation ranging from 5 to 35 mm, with the minimum in April and the maximum in August. Monthly corrections for wind-induced undercatch vary from 2 to 11 mm, or about 5-163%



increase of the gauge-measured amounts. The relative increase of monthly precipitation is much higher in the cold season (October to May) than in the warm season (June to September), due to the higher wind-induced undercatch for snow than for rain, and the smaller amount of absolute precipitation in the cold season. It is interesting to note the changes, due to bias corrections, in the precipitation regime at this site. Winter precipitation is doubled after the corrections because of strong winds and very low temperatures. The annual precipitation cycle continues to peak in summer, but significant bias corrections in winter reduce the winter-summer contrast.

At WCB, the monthly measured precipitation ranges from 12 to 55 mm, with the minimum in April and the maximum in August. Monthly corrections for wind-induced undercatch vary from 0 to 3 mm, or about 0-5% of the gauge-measured amounts. The monthly mean wind speeds are always below the lower threshold (1.2 m s$^{-1}$). The corrections for snowfall undercatch are negligible in the cold season (November to May) due to the small wind speed. The corrections for rain undercatch are relatively higher in the warm season (May to October) due to a constant correction factor of 5%. As a result, over-correction might occur for rainfall.

Monthly measured precipitation at WCF ranges from 2 to 55 mm, with the minimum in March and the maximum in August. Monthly corrections for wind-induced undercatch vary from 0 to 3 mm, or about 0-5 % increase of the gauge-measured amounts. The monthly mean wind speeds are always below the lower threshold (1.2 m s$^{-1}$). Similar to the WCB, the negligible corrections for snowfall undercatch are due to the low wind speeds (1.2 m s$^{-1}$).

The results for BERMS show monthly measured precipitation from 7 to 115 mm, with the minimum in March and the maximum in July. Monthly corrections for wind-induced undercatch vary from 0 to 6 mm, or about 0-5 % increase of the gauge-measured amounts. The monthly mean wind speeds are always below the lower threshold (1.2 m s$^{-1}$). The corrections for snowfall undercatch are negligible in the cold season (November to April).

Monthly measured precipitation at BC ranges from 6 to 70 mm, with the minimum in February and the maximum in July. Monthly corrections for wind-induced undercatch vary from 1 to 14 mm, or about 5-70% increase of the gauge-measured amounts. The relative increase of monthly precipitation is much higher (4-14 mm and 16-70%) in the cold season (October to April) than in the warm season (1-4 mm and 5-6%) (May to September), mainly due to the higher wind induced undercatch for snow than for rain and also smaller amount of absolute precipitation in the cold season.

Monthly measured precipitation at WNC ranges from 9 to 100 mm, with the minimum in January and the maximum in June. Monthly corrections for wind-induced undercatch vary from 3 to 30 mm, or about 5-123% increase of the gauge-measured amounts. The relative increase of monthly precipitation is much higher 23-123% in the cold season (October to May) than 5-23% in the warm season (June to September), mainly due to the higher wind-induced undercatch for snow than for rain. It is interesting to notice the changes, due to bias corrections, in precipitation regime at this site. Winter month precipitation has been doubled after the corrections because of high winds and very low temperatures.

For MC, monthly measured precipitation varies from 49 to 157 mm, with the minimum in February and the maximum in March. Monthly corrections for wind-induced undercatch ranges from 3 to 113 mm, or about 6-72%





increase of the gauge-measured amounts. The relative increase of monthly precipitation is much higher in the cold season (October to May) than in the warm season (June to September), as the result of higher wind induced undercatch for snow than for rain. Particularly, monthly corrections in March and April reach 113 mm and 93 mm, respectively. Bias-corrections significantly modify precipitation regime especially in March and June.

In comparison, wind-corrections for snowfall undercatch at the above sites can be grouped into two classes. For the sites WCB, WCF and BERMS with shielding of surrounding forest or brush, the corrections are negligible due to small wind speeds. For the open sites TVC, BC, WNC and MC, the corrections are high and vary among the sites, depending on regional climate factors such as air temperature, wind speed and snowfall percentage. For BC and WNC in the Prairie region, the rainfall-dominated precipitation regime remains the same after the bias-corrections. Significant

changes in precipitation regimes occur at TVC in the arctic region due to higher monthly mean correction factor 97% than that of 70% in cold-season at the Prairie sites. The most significant change in precipitation regime appears at MC in the Rocky Mountain Front Range due to the high snow percentage and high wind speed.

### 3.3 Annual precipitation and bias correction

An annual overview of the bias corrections is shown in Fig. 6, stratified by measured total precipitation, corrected rain,

corrected mixed precipitation and corrected snow ( $P_{obs}$, $P_{corr}^{R}$, $P_{obs}^{M}$, $P_{corr}^{S}$ ). The annual totals are shown in the left panels, and the corresponding contributions of the three precipitation types in the annual corrected totals are shown in the right panels.

For TVC, yearly measured precipitation ranges from 108 to 256 mm with a mean of 187 mm. Yearly corrected precipitation varies from 138 to 329 mm with a mean of 251 mm. The mean increase is 64 mm or 34.3%. The annual

mean contributions of bias corrections for rainfall, mixed one and snowfall are 7.5%, 20.8% and 71.7%, respectively. The bias corrections affect the interannual precipitation variability. For example, the ranking of 2011 precipitation changed from the fourth to the third.

For WCB, yearly measured precipitation ranges from 350 to 419 mm with a mean of 385 mm. Yearly corrected precipitation varies from 359 to 435 mm with a mean of 397 mm. The mean increase is 12 mm or about 3.1%. The

annual mean contributions of bias corrections for rainfall, mixed one and snowfall are 52.0%, 25.3% and 22.7%, respectively. The bias corrections do not change much of the precipitation pattern due to the high catch efficiency at this shielded site.

For WCF, yearly measured precipitation ranges from 241 to 303 mm with a mean of 270 mm. Yearly corrected precipitation varies from 247 to 314 mm with a mean of 279 mm for 2009-2011. The mean increase is 8 mm or 3.0%.

The annual mean contributions of bias corrections for rainfall, mixed one and snowfall are 84.8%, 15.2% and 0.0%, respectively. Yearly precipitation pattern remains the same after the corrections.

For BERMS, yearly measured precipitation ranges from 461 to 516 mm with a mean of 494 mm. Yearly corrected precipitation varies from 480 to 535 mm with a mean of 514 mm. The mean increase is 20 mm or 4%. The annual





mean contributions of bias corrections for rainfall, mixed one and snowfall are 88.2%, 11.7% and 0.1%, respectively. There is not much change in the interannual precipitation variation due to the high catch efficiency at this forest site.

Yearly measured precipitation at BC ranges from 308 to 475 mm with a mean of 372 mm. The snow percentage ranges from 11.0% to 39.0%. Yearly corrected precipitation varies from 345 to 549 mm with a mean of 432 mm. The mean increase is 60 mm or 15.1%. The annual mean contributions of bias corrections for rainfall, mixed one and snowfall are 27.1%, 29.6% and 43.3%, respectively. There is little change in the ranking of yearly precipitation due to the relative uniform and the low correction factor.

For WNC, yearly measured precipitation ranges from 325 to 555 mm with a mean of 427 mm. Yearly corrected precipitation varies from 437 to 667 mm with a mean of 553 mm. The mean increase is 126 mm, or 30.2%. The annual mean contributions of bias corrections for rainfall, mixed one and snowfall are 12.9%, 33.9% and 53.2%, respectively. The ranking of yearly precipitation has changed, and the observed minimum precipitation in 2009 became the sixth in yearly corrected precipitation.

At MC, yearly measured precipitation ranges from 847 to 1414 mm with a mean of 1087 mm. The snow percentage ranges from 57.3% to 76.1%. Yearly corrected precipitation varies from 970 to 2153 mm with a mean of 1471 mm. The mean increase is about 384 mm, or 32.6%. The annual mean contributions of bias corrections for rainfall, mixed one and snowfall are 3.6%, 18.3% and 78.1%, respectively. The ranking of yearly precipitation does not change much, but the increase in 2012 reached 739 mm due to heavy snowfall in spring.

In summary, the impact of the bias corrections on yearly precipitation is small at the sheltered sites with vegetation shielding (WCB, WCF and BERMS), ranging from 8 to 20 mm or 3-4%. At these sites, the bias-corrections do not change the precipitation patterns during the study periods. In contrast, the change in annual precipitation is larger at the open sites (TVC, BC, WNC and MC), ranging from 60 to 384 mm or 15-34%. In addition, the bias-corrections significantly alter the seasonal patterns of precipitation at windy sites with high snow percentage (>50%) (TVC and WNC). Similar features of precipitation bias-corrections have been reported over the high latitude regions (Yang and Ohata, 2001; Adam and Lettenmaier, 2003; Yang et al., 2005).

# 4 Discussion

Uncertainties exist in the analysis and intercomparison of precipitation gauge data and bias corrections. These include, for example, precipitation type determination including blowing snow events, observations, and calculations of mean wind speed and temperature for the time period of precipitation (Yang et al., 2014). Here we discuss some of the issues relevant to this study.

## 4.1 Light rain and snow

Analysis of the effect of filtering on rainfall measurements in Section 3.1 demonstrated that the filter algorithm works well in general. The resulting artifacts for light precipitation are influenced by noise status and rainfall characteristics (e.g. amount and intensity). However, the effect of filtering on snowfall measurements has not been verified.





Here we demonstrate the observed precipitation features (frequency) of rainfall and snowfall at the windy sites BC, WNC, TVC and MC in Fig. 7. The very light rainfall class (≤ 0.2 mm/30min) occurs most frequently, 36% to 56% of the time, with a mean of 43% at the former three sites. The corresponding snowfall frequency ranges from 34% to 67% with a mean of 62%. Whereas, it is the other way round at MC. Considering that wind speed is higher in the cold period than warm period, more light solid precipitation is prone to be obscured by diurnally fluctuating noise in the cold period, resulting in more filtering artifacts. In addition, since very high wind speed (> 10 m s$^{-1}$) events mainly concentrate in the cold period, for example, at WNC and TVC, bias correction with low catch efficiency will amplify the artifacts of filtering on light solid precipitation. Therefore, the effect of filtering on snowfall precipitation needs more attention in future.

## 4.2 Limitation of the wind speed - undercatch relationship

The results of this study may be limited by uncertainties and limitations in Eq. (1), which empirically relates the CE for solid precipitation to wind speed. One limitation is related to the observation interval of the DFIR with a manual gauge. Limited by the half-day or daily interval of manual observations of the DFIR at the Brats Lake experimental site in Saskatchewan, an adjusted double fenced Geonor gauge (Geonor-DF) hourly catch is indirectly referenced for the Geonor-SA catch. Recently data from DFIR with a Geonor gauge (Geonor-DFIR) are available from Brats Lake and they are useful to improve the catch-wind relationship for various automatic precipitation gauges. The correction function for the Geonor-DFIR vs the Geonor-SA has been derived, using the intercomparison data from Norway (Wolff et al., 2015). Its applicability to different regions and climatic conditions may need test and validation. We will consider testing various bias-correction methods in future analysis.

## 4.3 Effect of blowing snow and high winds

Blowing snow fluxes collected by precipitation gauges are called false precipitation measurements (Golubev, 1989). It is a challenge to quantify the effects of blowing snow on snowfall measurements because of a lack of necessary information. The magnitude of false precipitation is proportional to the intensity of the blowing snow and its duration. Based on field observations at a windy alpine location in the Colorado Front Range, Bardsley and Williams (1997) reported that blowing snow events often occur after storms with high wind speeds, over 20 m s$^{-1}$, and may introduce 50% overcatch over a winter season. Yang and Ohata (2001) found an association between higher wind speeds and higher snow measurement by the Tretyakov gauges at the windy and cold Tiksi and Dekson stations on the northern Siberian coast, perhaps resulting from snow blowing into the gauges. However, the quantity of overcatch by blowing snow might be overestimated due to the small mass concentrations of blowing snow at gauge height and the small terminal fall velocities (Pomeroy and Male, 1992). Blowing snow events were not identified in the data we used for this study. Figure 8 shows an example of plausible blowing snow event in 2011 at TVC. The continuously high wind speed events (Ws > 9 m s$^{-1}$) had lasted for more than one day around 26 February, and the corresponding bias adjustments, at the upper wind speed threshold, were at the adjustment maximum of 381%. If the measurement included "false precipitation" during blowing snow, the use of a bias correction will clearly magnify the forged precipitation. Unfortunately, information on blowing snow duration and intensity, critical to determining the blowing



snow flux and its impact on gauge observations in cold regions, is mostly unavailable (Sugiura et al., 2006; Sugiura et al., 2009). Because of the uncertainty in gauge performance during high wind conditions, it is difficult to assess the impact of blowing snow. More data collection using automated instruments in the cold regions and analyses of snowfall in higher wind conditions, including blowing snow events, are necessary.

## 5 Conclusions

This study applied a consistent filtering procedure for sub-hourly precipitation measurements from Geonor-SA gauges at seven experimental sites across different ecoclimatic regions in western Canada. It quantified the wind-induced biases in precipitation records over the period of 2006-2015, and documented variations and impacts of observation errors on precipitation patterns among the sites. The main conclusions are summarized as follows.

Two major types of noise are found in Geonor gauge measurements: short-term fluctuations with occasional spurious and irregular drops, and prolonged periods of decline. The former is related to temperature and wind speed changes, and the latter is caused by evaporation losses. The application of applied filtering procedure effectively removes these noises, as evidenced in the good agreement between rainfall measurements by the Geonor and TBRG gauges. However, the filter also introduces a few artifacts, removing some actual events and creating a few spurious ones, with an annual offset of rainfall less than 20 mm/yr.

Depending on the shielding of surrounding vegetation, the wind-corrections for snow undercatch vary significantly. For the sites WCB, WCF and BERMS with shielding by forest or brush, the bias-corrections are small. In contrast, corrections vary greatly among the open sites TVC, BC, WNC and MC, depending on regional climate factors such as air temperature, wind speed and snowfall percentage. On a yearly basis, the bias corrections increase total precipitation by 8 to 20 mm or 3-4% at the sites with vegetation shielding, and 60 to 384 mm or 15-34% at the open sites. Bias corrections also alter the seasonal patterns of precipitation at the windy, open sites where a large portion of precipitation falls as snow (WNC and TVC).

The effect of bias corrections on precipitation regime shows distinct regional features. For the rainfall dominated climatic regime in the Prairie region (e.g. BC and WNC), bias-corrections only slightly modify the seasonal patterns. For the arctic region (e.g. TVC), significant changes in precipitation regime occur after bias corrections due to higher gauge undercatch of snowfall in windy and cold climate conditions. For the Rocky Mountain Front Range (e.g. MC), the highest change among the sites is caused by high snow percentage and high wind speeds.

*Acknowledgment.* The following organizations provided data sets or funded field programs for data collection; the Natural Sciences and Engineering Research Council (NSERC) Changing Cold Regions Network, Environment Canada, Yukon Environment, Alberta Agriculture and Forestry and CFCAS IP3 Network; Alberta Environment, NSERC Discovery Grant, Canada-Alberta Water Supply Expansion Program, Royal Bank of Canada, Environment Canada Science Horizons Program, and Canadian Foundation for Climate and Atmospheric Science (DRI Network). We appreciate Bruce Johnson for supplying field photos of the gauges and Philip Harder's assistance for precipitation phase determination. The authors gratefully acknowledge the support from the Global Institute of Water Security,



University of Saskatchewan.

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



Table 1 Summary of site and instruments info, including heights of the precipitation gauges and wind sensors.

| Sites | Eco-regions | Instrument height (m) | | | Data period |
|---|---|---|---|---|---|
| | | Geonor | TBRG | Wind sensor | |
| Trail Valley Creek (TVC) | arctic tundra | 1.82, 1.9 | 1.25[c] | 1.82 | 2007-2013 |
| Wolf Creek / Buckbrush (WCB) | Sub-arctic shrub tundra | 1.75 | 4.76[c] | 4.76 | 2011-2012 |
| Wolf Creek / Forest (WCF) | Sub-arctic forest | 1.75 | 21.34[c] | 4.8 | 2009-2011 |
| BERMS / Old Jack Pine (BERMS) | Boreal forest | 1.82 | 5.0[a] | 2.5 | 2014-2015 |
| Brightwater Creek / Kenaston (BC) | Prairie/pasture | 1.5 | 0.3[a] | 2.0 | 2009-2015 |
| West Nose Creek / Woolliams Farm (WNC) | Prairie/cropland | 1.5 | 0.4[b] | 1.55 | 2006-2013 |
| Marmot Creek/ Fisera Ridge (MC) | Western Cordillera /Alpine tundra | 3.1/4.1[*] | 4.2[a] | 3.2/4.2[*] | 2011-2015 |

([a]Hydrological Services Tipping Bucket Rain Gauge (TB4); [b]Meteorological Service of Canada tipping bucket; [c]TE525MM gauge (Texas Electronics); [*]since February 8, 2013).



Table 2 Additional formulas for Eq. (2).

---

1. Diffusivity of water vapour in air, $D$ [m$^2$ s$^{-1}$]

$$D = 2.06 \times 10^{-5} \times \left( \frac{T_a + 273.15}{273.15} \right)^{1.75}$$

(Thorpe and Mason, 1966)

2. Thermal conductivity of air, $\lambda_t$ [J m$^{-1}$ s$^{-1}$ K$^{-1}$]

$$\lambda_t = 0.000063 \times (T_a + 273.15) + 0.00673$$

(List, 1949)

3. Sublimation & vaporisation, $L$ [J kg$^{-1}$]

$$L = \begin{cases} 1000 \times (2834.1 - 0.29T - 0.004T^2), & T < 0 \\ 1000 \times (2501 - 2.36T), & T \geq 0 \end{cases}$$

(Rogers and Yau, 1989)

4. Water vapour density, $\rho$ [kg m$^{-3}$]

$$\rho = \frac{m_w e}{RT},$$

$m_w$ : the molecular weight of water, 0.01801528 [kg mol$^{-1}$]

$R$: Universal Gas Constant, 8.31441 [J mol$^{-1}$ K$^{-1}$]

5. Vapour pressure, $e$ [kPa]

$$e = \frac{RH}{100} \times 0.611 e^{\frac{17.3T}{237.3+T}}$$

---





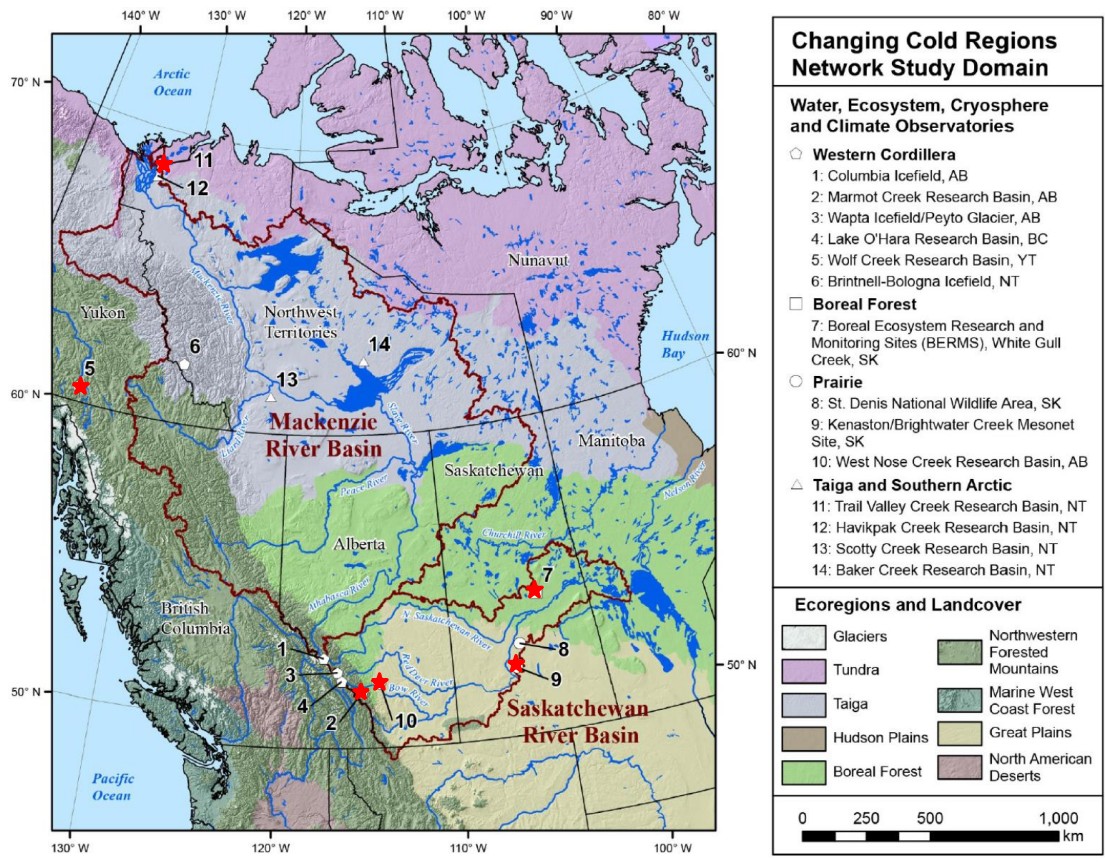

Figure 1 Locations of the selected five study sites (red stars) from the CCRN network with different ecoclimatic regions. Note that the number 5 includes two data sets. The base map of the interior of western and northern Canada is modified from DeBeer et al. (2015).



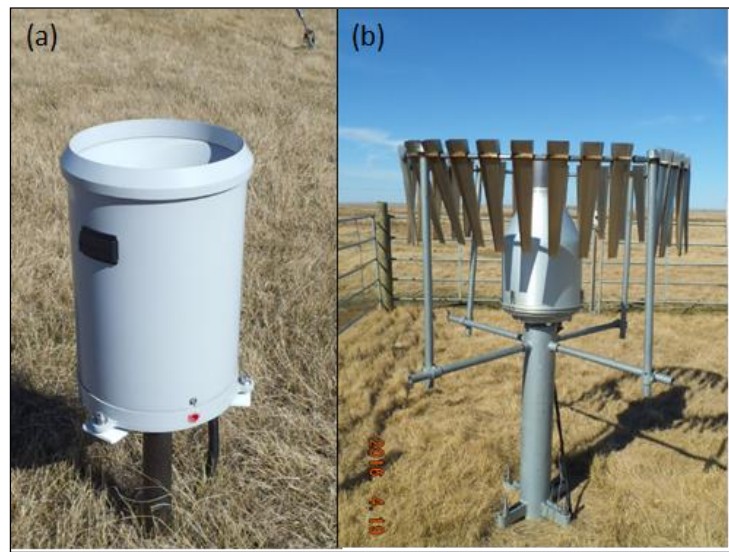

Figure 2 Two types of precipitation gauge used in this study. (a) Tipping bucket rain gauge; (b) the single Alter-shielded Geonor T200-B precipitation gauge.




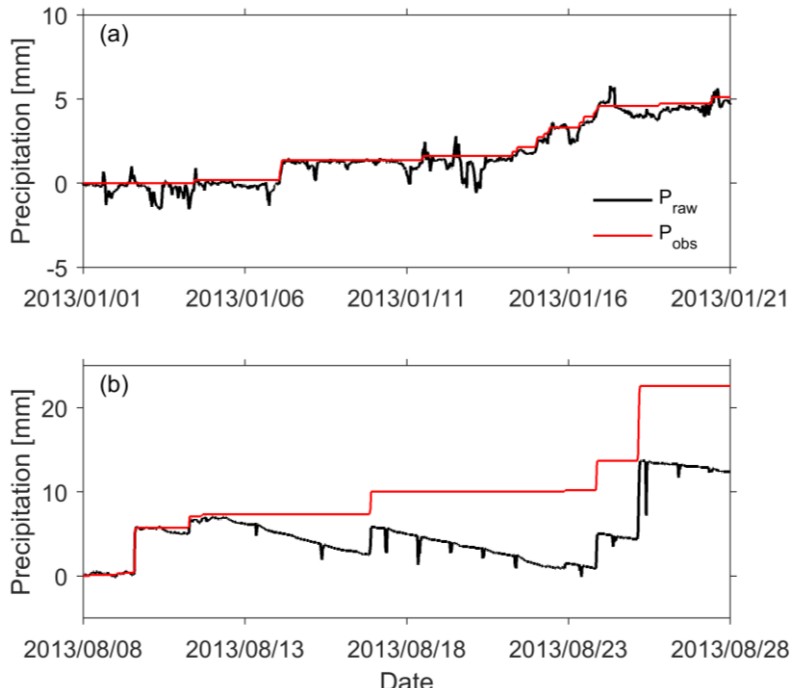

Figure 3 Examples of typical noise filtering for Geonor precipitation measurements ($P_{raw}$: raw data; $P_{obs}$: filtered data). (a) Diurnal drift, e.g. BC. (b) Evaporation caused drops, e.g. MC.





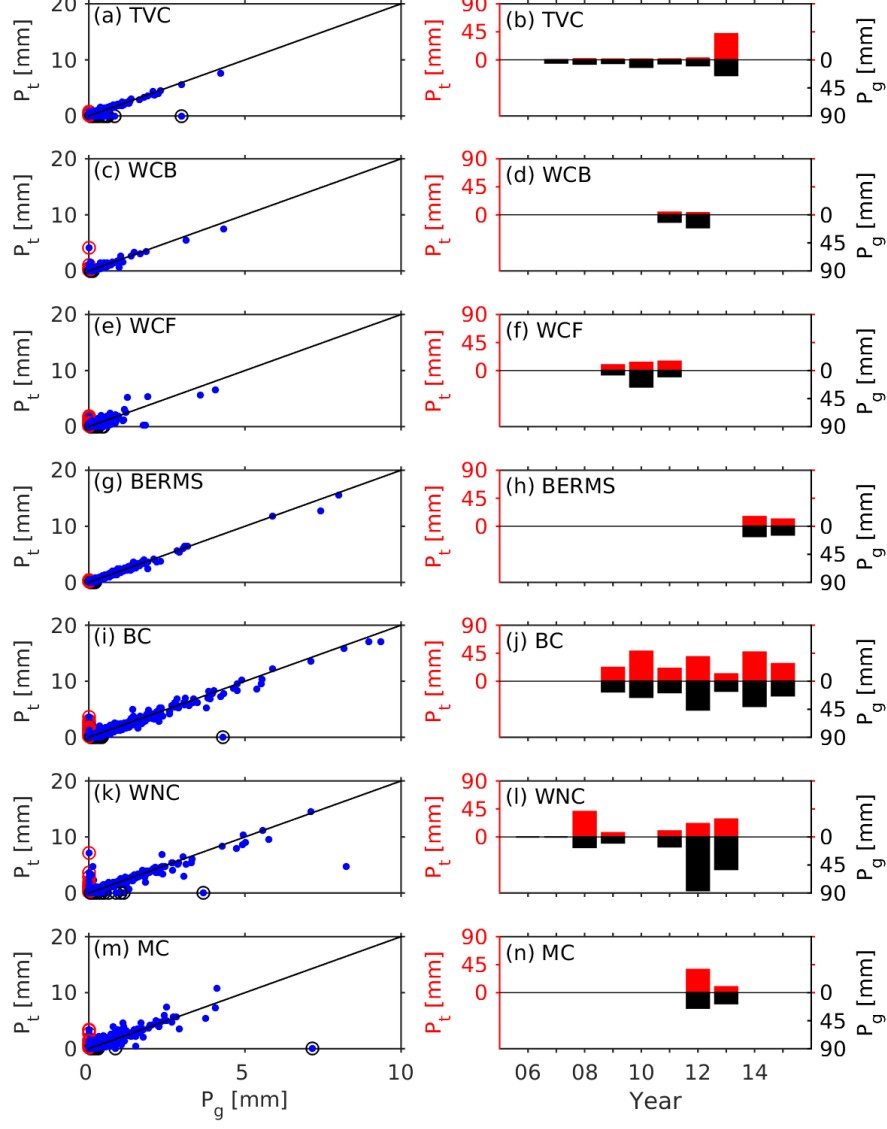

Figure 4 Comparison on rainfall rates from TBRG ($P_t$) and Geonor-SA ($P_g$) measurements at the sites (TVB, WCB, WCF, BERMS, BC, WNC and MC). The left panel compares the rainfall rates at the time scales of 30 minutes over the whole period. Blue points with red circle and black stand for two spatial cases: (1) $P_t > 0$, $P_g = 0$; (2) $P_t = 0$, $P_g > 0$. The right panel shows the yearly total amounts of "missing" precipitation at the two gauges. Note the different scales of y-axis in Fig. 4n.





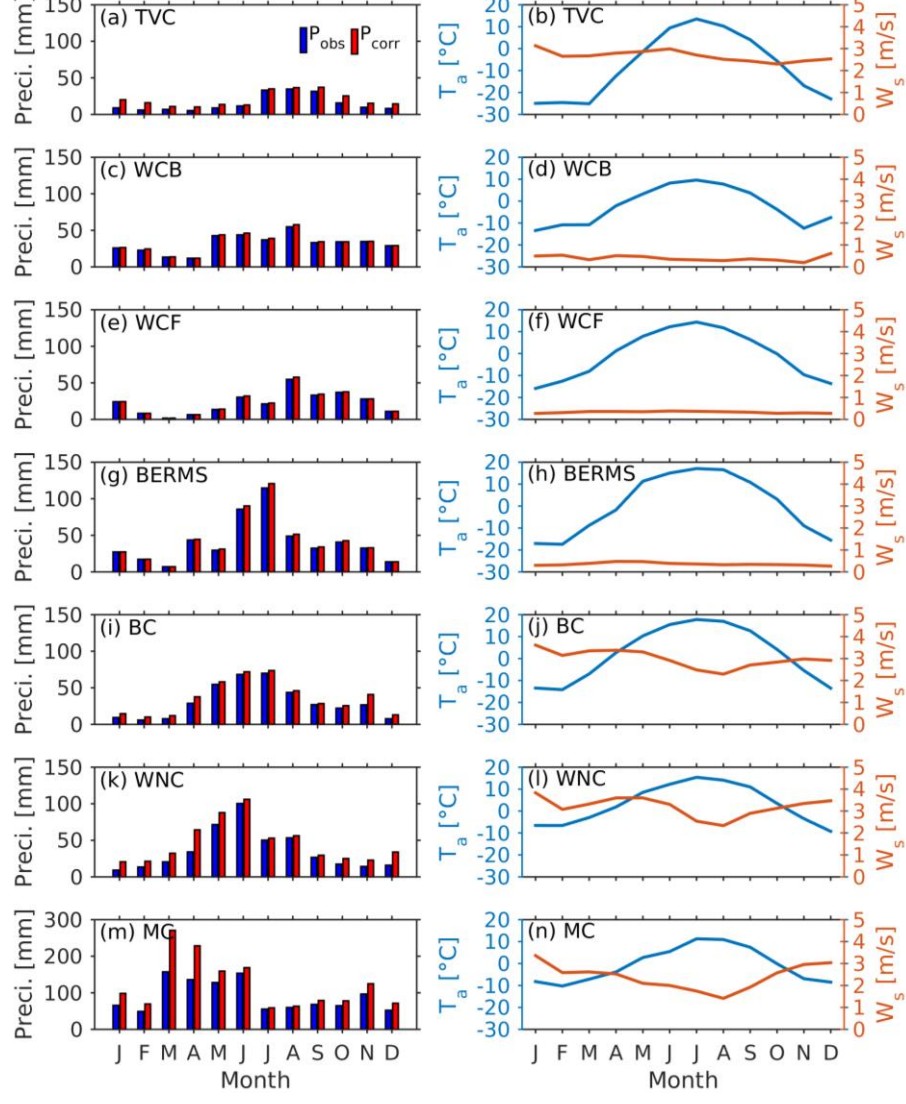

Figure 5 Comparison on precipitation correction in six different ecoclimatic regions. The left column plots (a), (c), (e), (g), (i), (k) and (m) compare the monthly averaged uncorrected and corrected precipitation rates ($P_{obs}$ & $P_{corr}$) at sites TVB, WCB, WCF, BERMS, BC, WNC and MC, respectively; and the right column plots (b), (d), (f), (h), (j), (l) and (n) shows the monthly mean air temperature ($T_a$) and gauge-height wind speed ($W_s$) at corresponding sites. Note the doubled scale of y-axis in (m).





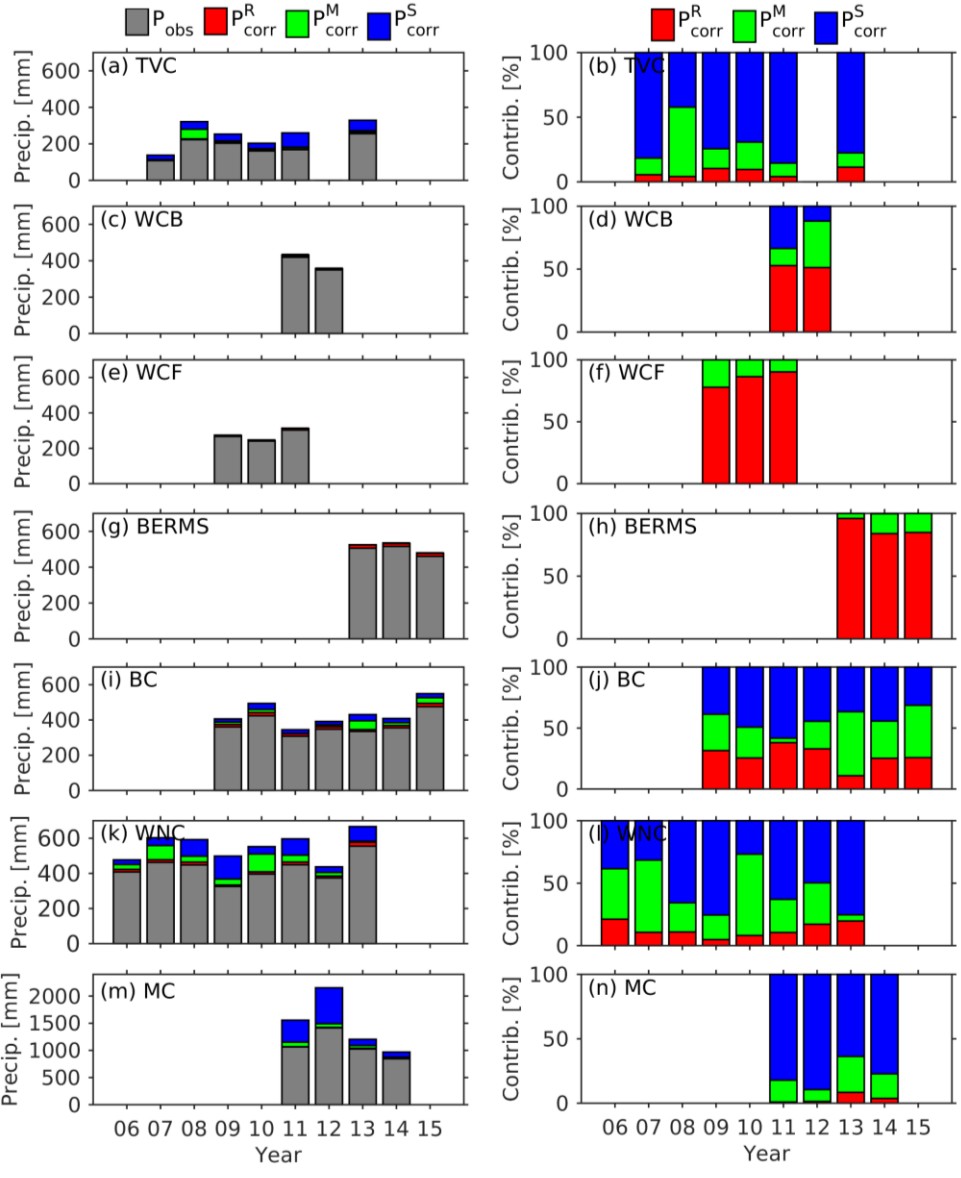

Figure 6 Variation of annual precipitation correction at the same sites as Fig. 5. The left column plots (a), (c), (e), (g), (i), (k) and (m) demonstrate the components of annual total precipitation: observed precipitation, corrected rain, corrected mixed precipitation and corrected snow ($P_{obs}$, $P_{corr}^{R}$, $P_{corr}^{M}$, $P_{corr}^{S}$) over the period of 2006 - 2015. The right column plots (b), (d), (f), (h), (j), (l) and (n) shows the annual contributions of the three precipitation types in the corrected precipitation.



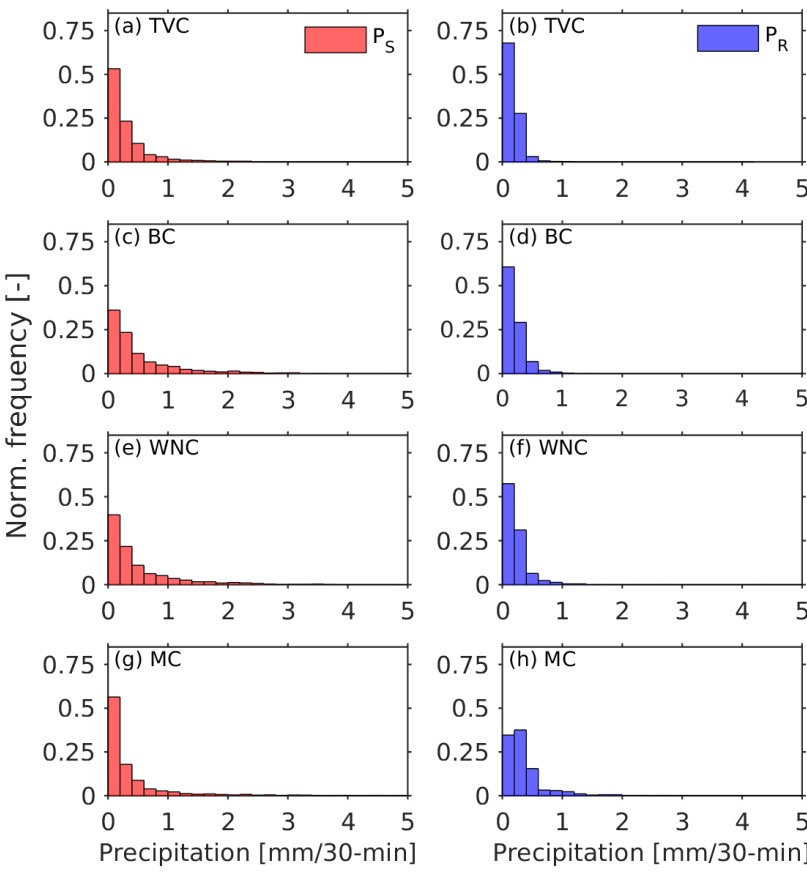

Figure 7 Histogram of 30 min rainfall ($P_R$: left column) and snow ($P_S$: right column) at four windy sites (TVC, BC, WNC and MC). For better visualization for light precipitation, low frequencies of the precipitation over 5 mm/30min were cut out.



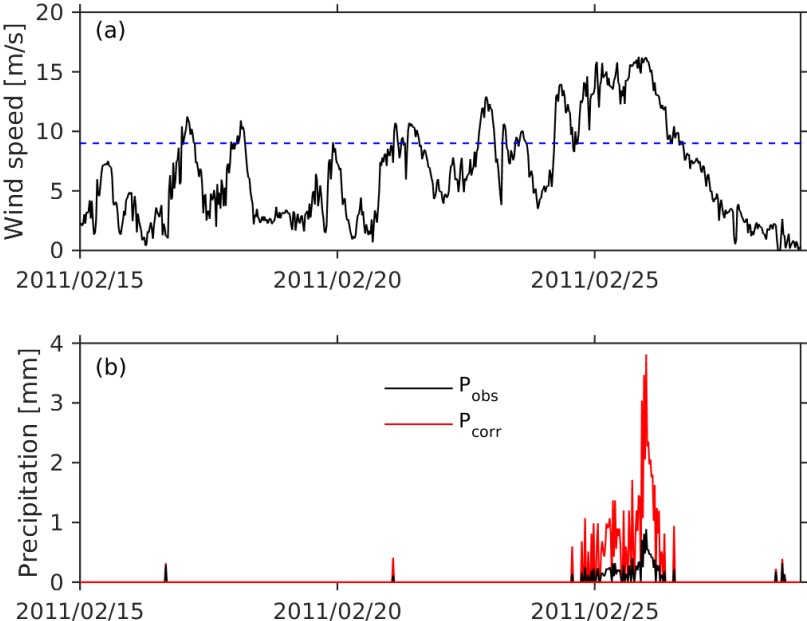

Figure 8. Example of plausible blowing snow event at TVC site. (a) Wind speed (dashed line: upper windspeed threshold for bias-correction), and (b) observed and bias-corrected snowfall over a period from 15 February to 1 March in 2011.