# Peer review of "Bias corrections of precipitation measurements across experimental sites in different ecoclimatic regions of western Canada"

_The Cryosphere, 2016_

## Referee Comment (RC1) · Anonymous Referee #1 · 4 Jul 2016

The paper is worthy of prompt publication. The methods, data and analysis support the Authors conclusion. The references are appropriate. The manuscript is well-written, logically organized and the figures and tables are appropriate, still some minor technical corrections are suggested in the following.

The Author investigate the impact of bias correction algorithms and quality control procedures on the observation of sub-hourly precipitation in Canada. In particular, they focus on the comparison of Geonor accumulating gauges and tipping bucket rain gauges measurements. The proposed quality control procedure aims to detect and properly remove not plausible measurements, while the correction for wind-induced

undercatch, which is tested but not developed in this paper, should correct for measurement biases, especially in the case of solid precipitation. The pros and cons of the quality control procedure are clearly described by the Authors. The impacts of the bias correction algorithm for different ecoclimatic regions in Canada is reported and the Authors carry out a detailed analysis for monthly, seasonal and annual time aggregations. In particular, they quantify the correction for the selected sites. In conclusion, the Authors stress the importance of (1) quality control for precipitation measurements and (2) the inclusion and testing of a correction for wind-induced undercatch of solid precipitation measurements within the post-processing routines of the raw measurements, especially in cold regions.

Minor corrections:

- page 7. line 13. "we have found t most events"

- page 15. Wahl et al. And Wolff et al. reference should be moved after Wang et al.

- Fig 4. "Note the different scales of y-axis in Fig 4n", still the scales are not different.

---

## Referee Comment (RC2) · Anonymous Referee #2 · 30 Aug 2016

The author does a nice job of explaining the importance of wind speed correction factors for solid precipitation and the importance of making these corrections. Some of the noise observed is probably due to the frequency of manual measurements and using the log law to extrapolate wind speed. The scientific approach is sound and the paper still makes a significant contribution of understanding catch efficiencies.

---

## Author Comment (AC2) · 30 Aug 2016

We thank the Referee for the comments. We agree that some noises might be operational artifacts.

---

## Author Response (AR1)

We would like to thank the Referee for the comments and suggestion, which are highly appreciated. The minor corrections are listed in the following.

1.page 7. line 13. "we have found t most events"
  Yes. "t" will be removed.

2.page 15. Wahl et al. And Wolff et al. reference should be moved after Wang et al.
  Agree.

3. Fig 4. "Note the different scales of y-axis in Fig 4n", still the scales are not different.
  Yes. It will be removed.

**Reply to**
We thank the Referee for the comments. We agree that some noises might be operational artifacts.

**Bias corrections of precipitation measurements across xperimental ites in different coclimatic egions 
[revised manuscript text omitted]

---

## Editor Decision (ED1)

**Minor comments for tc-2016-122:**

Page 3, line 16: BERMS seems to be part of another network, please clarify that in the text.

Page 4, lines 10-14: Please note already here, that you are using antifreeze in your Geonor gauges. It would also be helpful for the reader to give a specification of what kind of antifreeze and oil you are using.

Page 4, lines 22-29: I find it difficult to understand the "brute-force" filtering from the text.

*You are eliminating negative changes. Are small positive changes (under threshold) eliminated or combined until they reach threshold? If eliminated, what do you combine? I think that an extra figure and a few more sentences here would help the reader tremendously, especially as this is topic of very broad interest.*

Page 5, lines 27-28: add equation-symbol "f" to the sentence: ….between $T_i$ and the rainfall fraction $\boldsymbol{f_r}$ is then applied to separate….

*If you are not familiar with the method, it is difficult to understand that you are actually calculating directly the fraction of rainfall, as f is also widely used as "function of"*

Page 6, lines 21-25 and Page 12, lines 10-15: You are not mentioning high-frequent noise which may be present on Geonor measurements and is often caused by electro-magnetic-disturbances. I think that should be mentioned as it is another typical Geonor-noise which needs to be dealt with.

Page 6, line26/27: Any idea why you experience significant declines due to evaporation even with an oil layer? Significant declines occurred only at one station – what was different here? Please clarify if you have more information on the possible causes.

Page 11, line 17: The sentence "The correction function for the Geonor-DFIR vs the Geonor-SA has been derived…" seems a bit out of context. Did you want to give that as an example? Please state so. Great that you are considering effect of various bias-correction methods

Page 13, line 7: …Overcollection of **solid** precipitation

Page 13, line 24: ….and Brown, T.: **Multi**-variable evaluation of ….

Page 22, line 4: To help the readers understanding: Rewrite sentence starting with "Blue points…" Please move the color-explanation directly to the two cases, something like "Cases where only one of the gauges did measure precipitation are extra marked: red circles for Pt>0 and Pg=0 and black circles for Pt=0, Pg>0"

Page 24, line 5: Please specify that the annual contribution is given in percentage of the corrected precipitation amount, as it also could be the number of events

---

## Author Response (AR2)

We would like to thank the editor, Mareile Wolff, for the comments and suggestion, which are highly appreciated. Our responses to the comments are listed in the following.

*1. Page 3, line 16: BERMS seems to be part of another network, please clarify that in the text.*

Agreed. We have added, "(which originated during the BOReal Ecosystem Atmosphere Study and contributed to Fluxnet-Canada) …".

*2. Page 4, lines 10-14: Please note already here, that you are using antifreeze in your Geonor gauges. It would also be helpful for the reader to give a specification of what kind of antifreeze and oil you are using.*

Agreed. It is inserted in the revised manuscript.

*3. Page 4, lines 22-29: I find it difficult to understand the "brute-force" filtering from the text. You are eliminating negative changes. Are small positive changes (under threshold) eliminated or combined until they reach threshold? If eliminated, what do you combine? I think that an extra figure and a few more sentences here would help the reader tremendously, especially as this is topic of very broad interest.*

Agreed. We have added a more detailed description of the filtering algorithm.

*4. Page 5, lines 27-28: add equation-symbol "f" to the sentence: ….between Ti and the rainfall fraction fr is then applied to separate …. If you are not familiar with the method, it is difficult to understand that you are actually calculating directly the fraction of rainfall, as f is also widely used as "function of"*

Agreed. It is added in the revised manuscript.

*5. Page 6, lines 21-25 and Page 12, lines 10-15: You are not mentioning high-frequent noise which may be present on Geonor measurements and is often caused by electro-magnetic-disturbances. I think that should be mentioned as it is another typical Geonor-noise which needs to be dealt with.*

Agreed. We will mention it in the revised manuscript.

*6. Page 6, line26/27: Any idea why you experience significant declines due to evaporation even with an oil layer? Significant declines occurred only at one station – what was different here? Please clarify if you have more information on the possible causes.*

Here we confirmed that the significant declines at this site were caused by evaporation, when no oil was added.

7. Page 11, line 17: The sentence "The correction function for the Geonor-DFIR vs the Geonor-SA has been derived…" seems a bit out of context. Did you want to give that as an example? Please state so. Great that you are considering effect of various bias-correction methods

It is revised as "For example, the correction function for the Geonor-DFIR vs the Geonor-SA has been derived ..."

8. Page 13, line 7: …Overcollection of solid precipitation

Agreed.

9. Page 13, line 24: ….and Brown, T.: Multi-variable evaluation of …

Agreed.

10. Page 22, line 4: To help the readers understanding: Rewrite sentence starting with "Blue points…" Please move the color-explanation directly to the two cases, something like "Cases where only one of the gauges did measure precipitation are extra marked: red circles for Pt>0 and Pg=0 and black circles for Pt=0, Pg>0"

Agreed.

11. Page 24, line 5: Please specify that the annual contribution is given in percentage of the corrected precipitation amount, as it also could be the number of events

Agreed.

[revised manuscript text omitted]